# Influences of Cold Rolling and Aging on Microstructure and Property of CuCrSn Alloy

**DOI:** 10.3390/ma16103780

**Published:** 2023-05-17

**Authors:** Tao Chen, Qingke Zhang, Feng Liu, Xiaolong Feng, Cheng Xu, Zhenlun Song

**Affiliations:** 1Key Laboratory of Marine Materials and Related Technologies, Zhejiang Key Laboratory of Marine Materials and Protective Technologies, Ningbo Institute of Materials Technology and Engineering, Chinese Academy of Sciences, Ningbo 315201, China; chentao1@nimte.ac.cn (T.C.); xucheng@nimte.ac.cn (C.X.); songzhenlun@nimte.ac.cn (Z.S.); 2Ningbo Xingye Shengtai Group Co., Ltd., Ningbo 315336, China; liuf@cn-shine.com; 3Ningbo Kangqiang Electronics Co., Ltd., Ningbo 315105, China; fxl@kangqiang.com

**Keywords:** CuCrSn alloy, cold rolling, aging, tensile strength, conductivity

## Abstract

The CuCrSn alloy is promising as a high-strength and high-conductivity Cu alloy due to its relatively low smelting requirement. However, thus far investigations into the CuCrSn alloy are still quite lacking. In this study, the microstructure and properties of Cu-0.20Cr-0.25Sn (wt%) alloy specimens prepared under different rolling and aging combinations were comprehensively characterized, in order to reveal the effects of cold rolling and aging on properties of the CuCrSn. The results show that increasing the aging temperature from 400 °C to 450 °C can noticeably accelerate precipitation, and cold rolling before aging significantly increases its microhardness and promotes precipitation However, the deformation hardening is eliminated during the aging process, making the microhardness decrease monotonically when the aging temperature and the cold rolling ratio before aging are high. Performing cold rolling after aging can maximize precipitation strengthening and deformation strengthening, and the adverse impact on its conductivity is not serious. A tensile strength of 506.5 MPa and a conductivity of 70.33% IACS were obtained by such a treatment, whereas only the elongation decreases a little. Different strength-conductivity combinations of the CuCrSn alloy can be achieved through appropriate design of the aging and post-aging cold rolling conditions.

## 1. Introduction

The Cu-Cr series high-strength and high-conductivity alloys are typical precipitation-strengthened alloys with a high strength and excellent electrical conductivity, which have been widely applied in rail transit, electrical engineering, electronic packaging and some other fields [1,2,3,4,5]. The strengthening of the Cu-Cr series alloy is mainly due to the Cr-rich precipitates, and properties of the alloy can be further improved through adding more alloy elements such as Zr, Ti, Mg, Ag and rare earth elements [6,7,8,9,10,11]. Among them, the Cu-Cr-Zr alloy is the most widely used, while the Zr element can be easily burned during the smelting process and therefore vacuum smelting is required for it [12], resulting in a high cost and high price of the products, which limits further wide application of the Cu-Cr-Zr alloy.

Earlier investigations reveal that adding an appropriate amount of Sn into the Cu-Cr alloy can enhance the solid solution strengthening effect and improve the tensile strength of the alloy [13]. Meanwhile, Sn is relatively cheap and not easily burned. Therefore, vacuum melting is not required for the Cu-Cr-Sn alloy, and the manufacture costs are much lower, because vacuum smelting is usually at least three times more expensive than non-vacuum melting. Meanwhile, the raw material cost of Cu-Cr-Sn is also lower than that of Cu-Cr-Zr. However, thus far research reported on Cu-Cr-Sn alloys is still quite lacking compared with that on the other Cu-Cr series alloys, and the mechanical properties of Cu-Cr-Sn are still relatively poorer than those of the existing Cu-Cr-Zr alloys [2,4,14].

For the reasons above, in this study, a Cu-Cr-Sn alloy was designed and prepared, and then subjected to cold-rolling, solid solution and aging treatment under different conditions. The combination of deformation (rolling, drawing) and heat treatment (annealing, aging) is a common method to improve the mechanical properties of metals and has been widely used to optimize high-strength and high-conductivity Cu alloys [15,16]. The dislocations formed during the cold working process are considered to be promotive of precipitation [17,18], and cold rolling after aging can further improve strength [19]. The evolutions in grain structure, precipitates, and the mechanical and electrical properties of the Cu-Cr-Sn alloy during the cold rolling deformation and heat treatment processes were comprehensively characterized. The effects of cold rolling and aging on the microstructure as the well as properties of the Cu-Cr-Sn alloy were analyzed, and the relationship between the electrical properties, grain structure and precipitates were discussed. It is hoped that this study can provide some basis for optimizing the cold rolling aging parameters of the Cu-Cr-Sn alloy and thus improve the mechanical properties of the alloy.

## 2. Materials and Methods

### 2.1. Specimen Preparation

The composition of the designed alloy is Cu-0.20Cr-0.25Sn (wt%), and the alloy was smelted with pure Cu (≥99.99%), pure Sn (≥99.95%) and a Cu-10Cr (wt%) intermediate alloy in a muffle furnace under Ar protection. After the crucible was heated to 1250 °C, the metals were put into the crucible and left for 50 min and then casted in a graphite mold. Homogenization annealing of the alloy ingot was conducted in an Ar-protected furnace at 960 °C for 6 h, and cooled to room temperature with the furnace. An inductively coupled plasma emission spectrometer (ICP-OES, SPECTRO ARCOS II) was used to analyze the composition at different positions of the ingot, and the results showed that the average Cr content in the alloy was 0.201 wt%, and the average Sn content was 0.246 wt%, which is close to the design composition, indicating that quite accurate control of the alloy composition was achieved under the simple Ar-shielded melting condition in this study.

As the grains of the alloy after the homogenization annealing are coarse, the homogenization-annealed ingot was pre rolled from 10 mm to 2 mm, and then annealed at 960 °C for 1 h to refine the microstructure. Then, the alloy plate was further rolled to a thickness of 1.6 mm, 1 mm and 0.4 mm, respectively, and the corresponding rolling ratios were 20%, 50%, and 80%. The three rolling ratios were chosen to obtain a gradient and analyze the effects of rolling deformation. The sample (2 mm) after annealing was named CR0, and the 20%, 50% and 80% cold rolled samples were named CR1, CR2 and CR3, respectively. The production process of the Cu-Cr-Sn alloy in this study was relatively simple compared with that of the Cu-Cr-Zr alloy and can be conducted using the conventional Cu-alloy production line; thus, it is possible to introduce this Cu-Cr-Sn alloy into industrial processing. The specimens were aged at 400 °C and 450 °C for different times and then the mechanical properties and electrical conductivity of these specimens were characterized.

### 2.2. Microstructure Characterization

The samples for microstructure characterization were first mechanically ground, polished, and then etched with FeCl_3_ hydrochloric acid ethanol solution (ethanol 100 mL + FeCl_3_ 5 g + HCl 10 mL) for 25 s. The microstructure and phase of the alloy were characterized by Scanning Electron Microscopy (SEM, Sirion 200, FEI) and the precipitates were characterized by the Energy Dispersive Spectroscopy (EDS).

### 2.3. Microhardness and Conductivity Tests

To measure electrical conductivity and microhardness, the samples were firstly polished, and then the conductivity was measured using a Sigma 2008A digital conductivity meter at room temperature in air. After that, the microhardness was measured using a Vickers hardness tester (HV-1000) under a load of 200 g and a holding time of 10 s. For the microhardness and conductivity, each specimen was measured 5 times to obtain an average value, in order to minimize the errors.

### 2.4. Tensile Test and Fracture Surface Observation

According to the effect of aging conditions on properties of the cold-rolled Cu-0.2Cr-0.25Sn alloy, the Cu-Cr-Sn alloy with different cold rolling deformations were aged under their peak aging parameters: CR1 (400 °C, 2 h), CR2 (400 °C, 2 h) and CR3 (400 °C, 1.5 h), in order to obtain their highest strengths. The peak-aged specimens were named to be CR1-P, CR2-P, and CR3-P, respectively. The tensile and fracture behaviors of the specimens in peak aging states were studied. Additionally, the peak-aging parameter of the CR0 specimen was determined to be 450 °C, 2 h. The CR0 specimen, after peak aging, was cold-rolled from 2 mm to 0.4 mm, and named CR0-P. The effect of the cold-rolling and aging sequence on the electrical and mechanical properties of the alloy was studied through comparing the CR3-P and CR0-P specimens. The processing sequences of all the specimens are presented in Table 1.

To conduct the tensile test, the cold-rolled plates were wire cut into bone-shaped tensile specimens, with a total length of 80 mm, a gauge distance of 30 mm, and a transition arc radius of 5 mm. The side surfaces of the tensile specimens were carefully ground to remove the cutting marks. The tensile test was conducted on a universal testing machine (Zwick/Roell Z030) under a tensile speed of 2 mm/min at 20 °C in air, and the fracture surfaces were observed using SEM. Three samples were tested for each kind of specimen.

## 3. Results and Discussion

### 3.1. Microstructure and Properties of the Cold-Rolled CuCrSn Alloy

The microstructures of the Cu-0.2Cr-0.25Sn alloy plates annealed (solid solution treated) at 960 °C for different times and then cold-rolled to different thicknesses are shown in Figure 1. As can be seen in Figure 1a, the texture formed during the pre-rolling process disappears after the solid solution treatment, and the obvious annealing twin structure appears. Meanwhile, there are no obvious precipitation particles in the matrix, indicating that the Cr and Sn alloy elements were fully dissolved. From Figure 1b–d, it can be seen that the microstructure of the alloy changes with an increasing rolling ratio. Due to the low rolling ratio, the CR1 alloy almost retains the high-temperature-annealing recrystallization structure (see Figure 1b). As the rolling ratio increases, an obvious strip-like texture appear in the alloy, which is more obvious in CR3, and the grain boundary density in it is higher compared with CR2. Additionally, the etched surface is coarser at a higher rolling ratio, because the dislocations promote a corrosion reaction [20,21]. Some pores can be observed on the surface of CR3, despite the fact that all the specimens were prepared from the same ingot, and there were no pores before surface corrosion of the specimens. As high-density dislocation was introduced into the alloy during the cold-rolling process, which promoted corrosion, and the rolling ratio of CR3 was high, more corrosion pores can be observed on the CR3.

The microhardness and conductivity of the CR0, CR1, CR2 and CR3 specimens are shown in Figure 2. The microhardness of the CR0 alloy was very low, because there was no deformation strengthening or precipitation strengthening in this specimen. With an increasing cold-rolling ratio, the microhardness increased continuously. In contrast, as the Cr and Sn dissolved into the Cu matrix, the conductivity of CR0 was relatively low and further deceased during the rolling process, because the defect density in the recrystallized Cu alloy was low. The cold rolling introduces defects such as dislocations because high density dislocations become intertwined with each other, forming a cutting order and resulting in a significant dislocation strengthening effect [22,23]. Additionally, the high-density dislocations have a certain degree of a scattering effect on the free electrons in the Cu matrix and thus decrease the conductivity.

### 3.2. Properties of the Cold-Rolled CuCrSn Alloy after Different Aging Conditions

The evolutions in microhardness and conductivity of the CuCrSn alloy during aging at 400 °C are shown in Figure 3. According to Figure 3a, for all the specimens, the microhardness increased firstly and then decreased with increasing aging time. For the CR0 alloy, it was found that the rate of increase of the microhardness and conductivity was much lower than the other three specimens, and the highest (peak) values of the microhardness and conductivity were also much lower. One reason for that is that the CR0 alloy had not undergone cold rolling and lacked the deformation defects to promote precipitation, and the other reason is that the precipitation power may have been insufficient at the aging temperature of 400 °C. As a result, the CR0 alloy was in an underaged state at 400 °C and thus its microhardness and conductivity were much lower.

For the cold rolled specimens, the time for the alloy to reach the peak aging state decreased as the cold-rolling ratio increased. Microhardness of the CR3 alloy reached its peak after aging for only 1.5 h, indicating that cold deformation before aging can accelerate or promote precipitation. With further prolongation of the aging time, the precipitates gradually grew and the alloy became overaged., i.e., the precipitation strengthening effect was weakened and the microhardness gradually decreased. The conductivity firstly increased rapidly and then remained stable with increasing aging time, as presented in Figure 3b. At the early stage of aging, the conductivity increased rapidly due to the rapid precipitation of alloy elements dissolved in the matrix. The larger the cold-rolling ratio before aging, the higher increase rate of the conductivity, which demonstrates that cold rolling can promote precipitation because the internal defects in the grains can accelerate diffusion and promote the nucleation of the second phase [24,25]. After that, only a little of the alloy element will be further precipitated, so the conductivity tends to be stable. The stable conductivity of the CR1, CR2 and CR3 alloys are all around 74% IACS. It should be noted that the solubility of alloy elements is higher at a higher temperature, so increasing the aging temperature is adverse for conductivity.

Figure 4 shows the evolution in properties of the CuCrSn alloy during aging at 450 °C. From Figure 4a, it can be seen that the rate of increase for the microhardness of the CR0 alloy increased significantly compared with the alloy aged at 400 °C, because the higher aging temperature increased the precipitation power of the second phase and promoted precipitation. The time for the four specimens to reach peak aging was shortened compared with that in Figure 3, especially for the CR3 alloy, for which the microhardness decreased monotonically with increasing aging time. As the cold-rolling ratio of the CR3 was relatively high, its initial microhardness was also high due to severe strain hardening. During the aging process, the strain hardening decreased sharply, and precipitation hardening increased firstly and then decreased. Once the precipitation hardening could not make up the decrease in the strain hardening, a monotonic decrease in microhardness occurred. Additionally, the peak hardness of the specimens aged at 450 °C were lower than those aged at 400 °C, indicating that an aging temperature of 400 °C is more than enough for the cold-rolled specimens. The variation in the conductivity shown in Figure 4b is consistent with that shown in Figure 4a, in which the conductivity of the CR0 alloy increased significantly. The higher aging temperature increases the precipitation driving force of the alloy and accelerates the precipitation rate. The conductivity of all the 4 specimens were higher than 70% IACS, with some at about 75% IACS.

### 3.3. Tensile Behavior of the Peak-Aged CuCrSn Alloy

The Cu-Cr-Sn alloy specimens with different cold-rolling ratios were aged under their peak-aging parameters: CR1 (400 °C, 2 h), CR2 (400 °C, 2 h) and CR3 (400 °C, 1.5 h), and the aged specimens were named CR1-P, CR2-P, and CR3-P, respectively. The engineering tensile stress–strain curves of the CR1-P, CR2-P and CR3-P specimens are shown in Figure 5, and their strengths are presented in Figure 6. It can be found that with an increasing cold-rolling ratio before aging, the yield strength and tensile strength increased obviously. One reason for this is that dislocations in the cold-rolled specimens can promote the nucleation of the precipitates, and thus a higher density and finer precipitation particles can be formed [17,18]; thus, the CR3-P specimen has much higher strength. Part of the deformation strengthening might be maintained, but should not be significant at the aging temperature of 400 °C [26,27]. For CR0-P, CR1-P and CR2-P, the difference in solid solution strengthening and deformation strengthening should be small, and there may be a difference in the grain size of the Cu matrix and a difference in precipitation. Furthermore, as fine-grain strengthening should improve the strength whilst increasing the elongation, but the elongation decreases with increasing strength in Figure 5, it can be predicted that the increase in strength is mainly attributed to precipitation strengthening rather than fine grain strengthening.

The microscopic fracture surfaces of the peak aged specimens are shown in Figure 7. For all three specimens, dimples were formed on the fracture surface, demonstrating that the specimens fracture in a ductile mode. For the CR1-P specimen, the dimples were relatively large in size and depth, indicating a better plasticity. With higher cold-rolling deformation before aging, the dimples become smaller and shallower, corresponding to a higher strength and lower elongation. The fracture surfaces correspond well to the tensile curves and the strength.

### 3.4. Properties of the CuCrSn Alloy Cold-Rolled after Aging

The deformation-aging process has a significant influence on the microstructure and properties of the Cu alloys. As the tensile strength of the CR3-P is only 455.5 MPa, the “solid solution-peak aging-cold rolling” treatment was conducted to further improve the strength of the Cu-Cr-Sn alloy. The CR0-P specimen was pre-rolled from 10 mm to 2 mm, solid-solution-treated at 960 °C for 1 h, aged at 450 °C for 2 h and then cold-rolled to 0.4 mm. Tensile curves of the CR0-P and CR3-P specimens are shown in Figure 8a, from which one can find that the tensile strength of the CR0-P specimen is over 500 MPa, much higher even than that of CR3-P, but its elongation decreases sharply. The fracture surface of the CR0-P also revealed a decrease in ductility. As shown in Figure 8b, the fracture surface is flat; only a few shallow dimples can be observed, while there is still little cleavage on the fracture surface., i.e., it is still not brittle.

The microstructure of the CR0-P specimen and results of the EDS analysis on particles from this specimen are shown in Figure 9. In the microstructure image, rolling deformation bands can be observed, and dispersed fine white particles exist in the Cu matrix, as indicated by the arrows. The deformation bands are formed due to the difference in Cu matrix grain orientation and in turn the plastic deformation degree. The EDS analysis results reveal that these precipitates are Cr-rich particles, which are formed during the aging process and act as the strengthening phase, while cold rolling after aging further improves its strength. However, the size of the Cr-rich phase is still high, and may be refined to further optimize the strengthening effect. The properties of the CR0-P and CR3-P specimens are summarized in Figure 10, in which it is obvious that the yield strength, tensile strength and microhardness of the CR0-P specimen are higher; only the conductivity is a little lower, but still higher than 70%.

According to the results above, the microhardness of the solid-solution-treated Cu-0.2Cr-0.25Sn specimen increased by 54 HV after peak aging, from 59 HV to 113 HV. After a further 80% cold rolling, the microhardness increased again to 170 HV (CR0-P). For this specimen, the strengthening can be attribute to the combined effects of precipitation strengthening and deformation strengthening. With another processing method, the microhardness of the Cu-Cr-Sn specimen increased from 59 HV in the solid solution state to 152 HV in the 80% cold-rolled state before aging. Then, the microhardness increased to 165 HV after the peak-aging treatment, which indicates that under the “solid solution-cold rolling-aging” process, the contribution of deformation strengthening is not significant, because aging will eliminate the dislocations introduced during the cold rolling process [27,28]. Although cold rolling decreases the conductivity of the CR0-P specimen by a few percentages of IACS, it is still higher than 70% IACS. Through comparing the properties of the CR3-P and CR0-P specimens, it can be concluded that adjusting the sequences of cold rolling and aging can improve the strength of a Cu-Cr-Sn alloy whilst it maintains a superior conductivity. Moreover, the production process of the Cu-Cr-Sn alloy in this study is relatively simple and can be conducted using the conventional Cu alloy production line. Therefore, it is possible to introduce this method into industrial processing.

## 4. Conclusions

The influence of the cold-rolling ratio and aging parameters on the microstructure and property of a Cu-0.2Cr-0.25Sn alloy were investigated in this study. Based on the results and discussions, the following conclusions can be drawn:(1)Cold rolling can change the coarse recrystallization structure of the alloy, and forms a deformation texture and bands with severe plastic deformation. A high density of dislocations were introduced during the cold rolling process, which significantly increased the microhardness but decreased the conductivity by a little.(2)The microhardness increases firstly and then decreases during the aging process, while the conductivity increases rapidly at the early aging stage and then becomes stable. Cold rolling before aging can accelerate the precipitation process, and the peak microhardness is also higher for the specimen with a higher cold-rolling ratio before aging. The most suitable aging temperature is related to the cold-rolling ratio before aging. Improving the aging temperature shortens the peak aging time.(3)Through aging firstly and then conducting cold rolling, the strength of the Cu-0.2Cr-0.25Sn alloy can be further improved, because the precipitation strengthening and deformation strengthening are both significant in this condition, and the conductivity can still maintain a relatively high level. A tensile strength of 506.5 MPa and a conductivity of 70.33% IACS were obtained for the specimen with the processes of annealing at 960 °C for 1 h, aging at 450 °C for 2 h and then 80% cold rolling.

## Figures and Tables

**Figure 1 materials-16-03780-f001:**
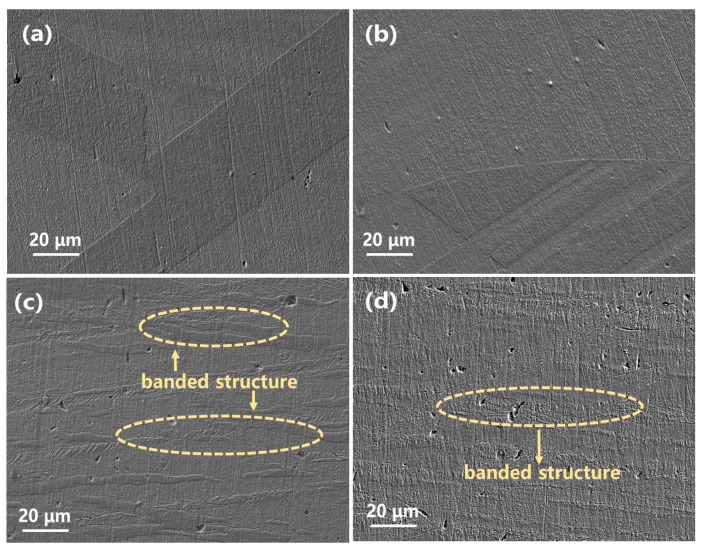
Microstructures of the Cu-0.2Cr-0.25Sn alloy specimens of different processing states: (**a**) CR0, (**b**) CR1, (**c**) CR2, (**d**) CR3.

**Figure 2 materials-16-03780-f002:**
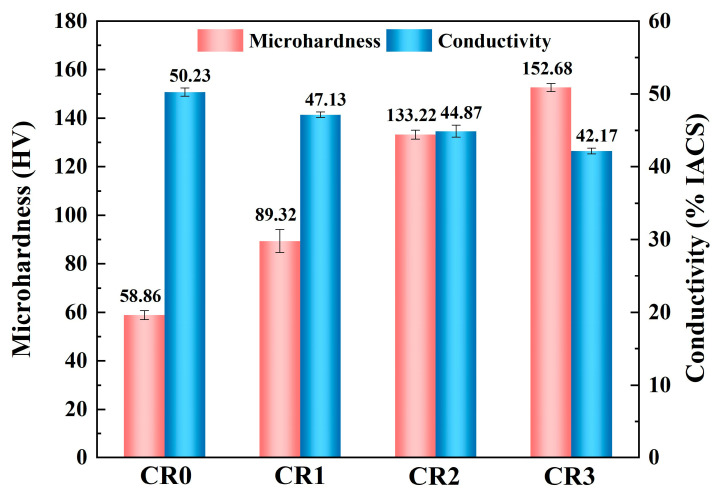
Microhardness and electrical conductivity of the Cu-0.2Cr-0.25Sn alloy specimens of different processing states.

**Figure 3 materials-16-03780-f003:**
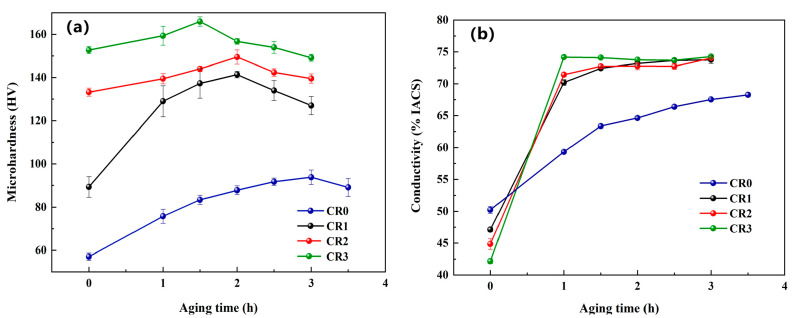
Evolutions in properties of the CuCrSn alloy during the aging at 400 °C: (**a**) microhardness, (**b**) conductivity.

**Figure 4 materials-16-03780-f004:**
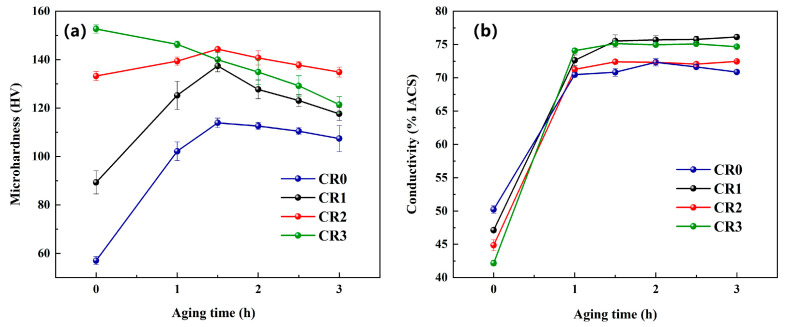
Evolutions in properties of the CuCrSn alloy during the aging at 450 °C: (**a**) microhardness, (**b**) conductivity.

**Figure 5 materials-16-03780-f005:**
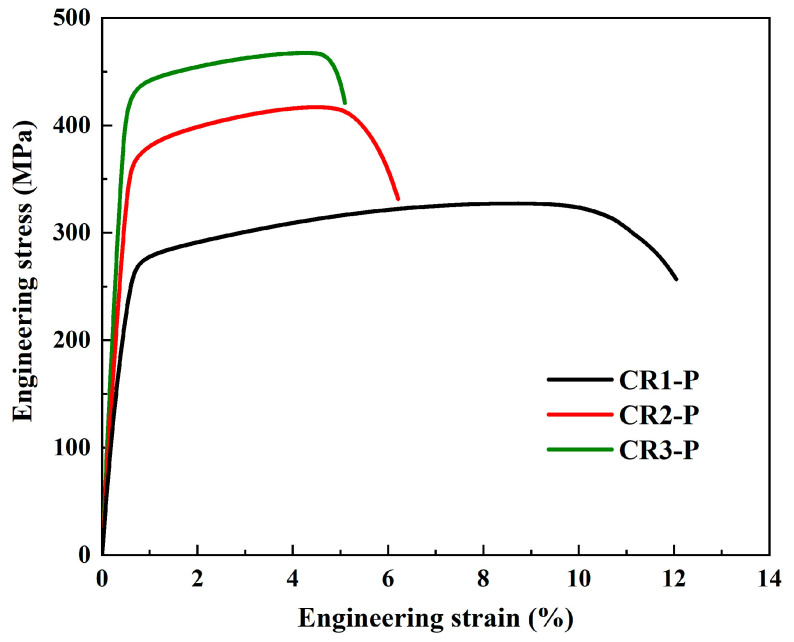
Tensile stress–strain curves of the specimens under their peak aging states.

**Figure 6 materials-16-03780-f006:**
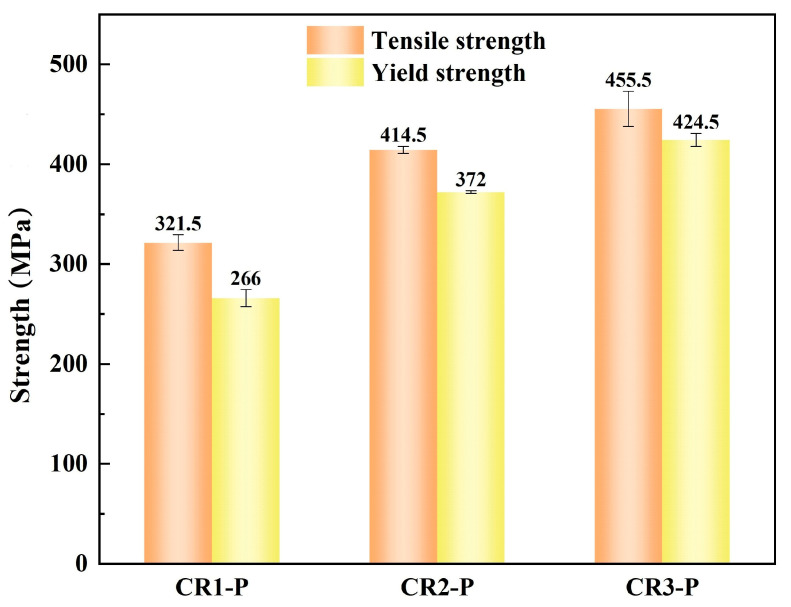
Strength of the specimens under their peak aging states.

**Figure 7 materials-16-03780-f007:**
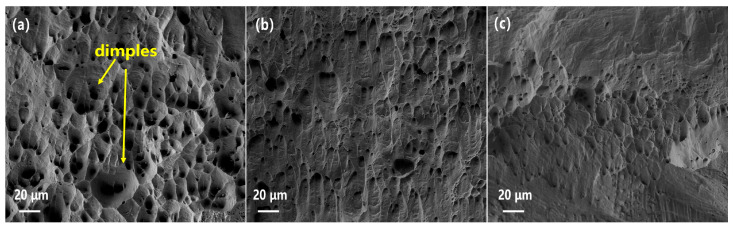
Fracture surfaces of the specimens under their peak aging states: (**a**) CR1-P, (**b**) CR2-P, (**c**) CR3-P.

**Figure 8 materials-16-03780-f008:**
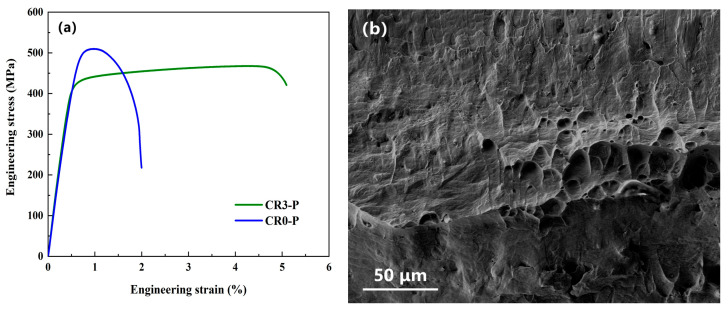
(**a**) Tensile curves of the CR3-P and CR0-P specimens, (**b**) fracture morphology of the CR0-P specimen.

**Figure 9 materials-16-03780-f009:**
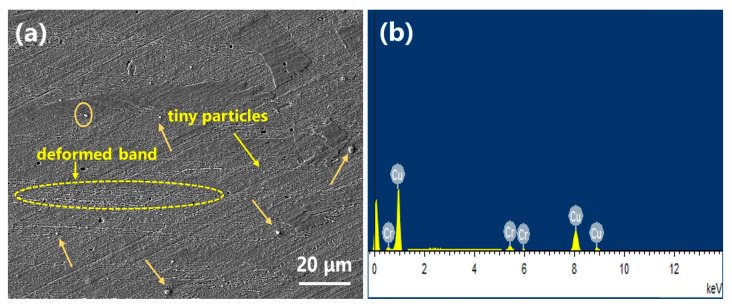
(**a**) microstructure and (**b**) EDS analysis result of the CR0-P specimen.

**Figure 10 materials-16-03780-f010:**
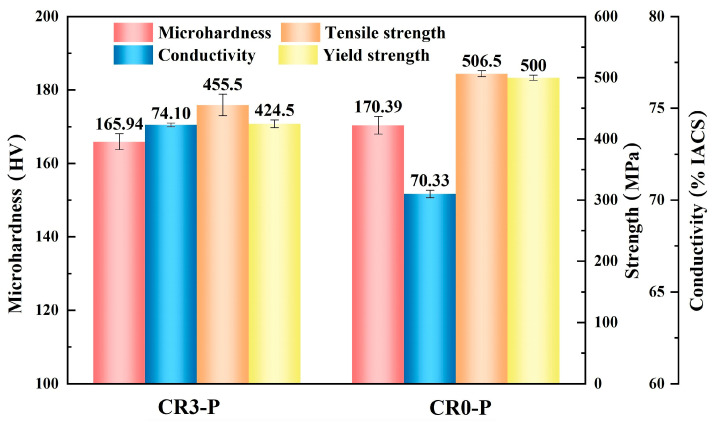
Properties of the CR0-P and CR3-P specimens.

**Table 1 materials-16-03780-t001:** Processing sequences and serial numbers of the specimens.

Serial Number	Pre Rolling	Annealing	First Rolling	Aging	Further Rolling
CR0	10 mm to 2 mm	960 °C, 1 h	/	400 °C and 450 °C for different time	/
CR0-P			/	450 °C, 2 h	2 mm to 0.4 mm
CR1			2 mm to 1.6 mm	400 °C and 450 °C for different time	/
CR1-P			2 mm to 1.6 mm	400 °C, 2 h	/
CR2			2 mm to 1.0 mm	400 °C and 450 °C for different time	/
CR2-P			2 mm to 1.0 mm	400 °C, 2 h	/
CR3			2 mm to 0.4 mm	400 °C and 450 °C for different time	/
CR3-P			2 mm to 0.4 mm	400 °C, 1.5 h	/

## Data Availability

The data presented in this study are available on request from the corresponding author.

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
