# Peer review of "Influences of Cold Rolling and Aging on Microstructure and Property of CuCrSn Alloy"

_materials, 2023, doi:10.3390/ma16103780_

Round 1

Reviewer 1 Report

  1. Explain the reasons behind your selection of the rolling ratio given in Table 1.
  2. The quality of figures are poor and suggested to improve the resolution of all figures.
  3. Why microhardness at 400 and 450 oC  for CR0 is distinct. This requires clarification.
  4. In Page 6, 185 line mentioned that “ higher density and finer precipitation particles can be formed thus the CR3-P specimen has much higher strength”.  Need proof for the above statement.
  5. In Page 6, 188 line mentioned that “As the elongation decreases with increasing strength, it can be predicated that the increase in strength mainly attributes to the precipitation strengthening rather than fine grain strengthening”. Required proper proof for the above statement.
  6. Need more explanation about the Fig 9.

Author Response

  1. Explain the reasons behind your selection of the rolling ratio given in Table 1.

Re: Thanks for the reminding. The three rolling ratios (20%, 50% and 80%) were chosen to get a gradient and analyze the effects of rolling deformation.

  1. The quality of figures are poor and suggested to improve the resolution of all figures.

Re: Thanks for the suggestion. All the figures have been replaced by the high resolution images.

  1. Why microhardness at 400 and 450 ºC for CR0 is distinct. This requires clarification.

Re: The initial microhardness of the CR0 before aging is 58.86 HV, which are the same in Fig. 3 and Fig. 4, only the vertical coordinate ranges in the two figures are a little different.

  1. In Page 6, 185 line mentioned that “ higher density and finer precipitation particles can be formed thus the CR3-P specimen has much higher strength”. Need proof for the above statement.

Re: Thanks for the reminding. Some references have been added as basis for the statement.

  1. In Page 6, 188 line mentioned that “As the elongation decreases with increasing strength, it can be predicated that the increase in strength mainly attributes to the precipitation strengthening rather than fine grain strengthening”. Required proper proof for the above statement.

Re: Thanks for the reminding. It is a discussion based on the effects of precipitation strengthening and fine grain strengthening on the elongation. For CR0-P, CR1-P and CR2-P, there may be difference in grain size of the Cu matrix and difference in precipitation. As the fine grain strengthening will improve the strength and meanwhile increase the elongation, while the elongation decreases with increasing strength in Fig. 5, it is predicated that the increase in strength mainly attributes to the precipitation strengthening rather than fine grain strengthening. That has been explained in more detail in the revised manuscript.

  1. Need more explanation about the Fig. 9.

Re: Thanks for the kind suggestion. The formation mechanisms of the rolled deformation bands and effects of the Cr-rich phase have been supplemented to the explanation of Fig. 9.

Reviewer 2 Report

The manuscript is about the influences of cold rolling and aging on microstructure and property of CuCrSn alloy. Language is enough. Please consider the following comments below:

(1) The introduction part is not enough. It needs to be expanded. Additionally, please add more information about current studies on aging and rolling process applications applied to other Cu alloys.

(2) The study does not discuss the cost of producing the CuCrSn alloy compared to other Cu alloys or materials. A cost analysis would provide useful information for evaluating the economic viability of the CuCrSn alloy for different applications.

(3) Fig.3, fig.4, fig.8a, and fig.9b are blurred. Please improve the resolution of the figures.

(4) why did not authors apply XRD analysis to determine phases in the microstructure?

XRD analysis can provide more detailed information about the crystal structure and phase composition of the material, which can be useful for understanding the underlying mechanisms responsible for the observed mechanical properties. Therefore, if detailed phase analysis is required, XRD analysis may be necessary.

(5) The study does not mention the number of specimens used in each treatment group.

(6) the testing temperature used for the tensile tests can be informed in related method section.

(7) CR3 has the highest mechanical properties. However, in Figure 1, it is clearly seen that CR3 includes more pores than other SEM images. How do the authors explain the contradiction?

Author Response

The manuscript is about the influences of cold rolling and aging on microstructure and property of CuCrSn alloy. Language is enough. Please consider the following comments below:

 (1) The introduction part is not enough. It needs to be expanded. Additionally, please add more information about current studies on aging and rolling process applications applied to other Cu alloys.

Re: Thanks for the kind reminding. More introduction on application of aging and rolling to the Cu alloys has been added into the revised manuscript.

(2) The study does not discuss the cost of producing the CuCrSn alloy compared to other Cu alloys or materials. A cost analysis would provide useful information for evaluating the economic viability of the CuCrSn alloy for different applications.

Re: Thanks for the suggestion. The difference in cost for production of the Cu-Cr-Sn and Cu-Cr-Zr alloys is mainly resulted from the difference of vacuum melting and gas shield smelting. Usually, vacuum smelting is at least three times more expensive than non vacuum melting.

(3) Fig.3, fig.4, fig.8a, and fig.9b are blurred. Please improve the resolution of the figures.

Re: Thanks for the suggestion. All the figures have been replaced by the high resolution images.

(4) why did not authors apply XRD analysis to determine phases in the microstructure?

XRD analysis can provide more detailed information about the crystal structure and phase composition of the material, which can be useful for understanding the underlying mechanisms responsible for the observed mechanical properties. Therefore, if detailed phase analysis is required, XRD analysis may be necessary.

Re: Thanks for the suggestion. In fact, there are only Cu-rich matrix and Cr-rich precipitates in the Cu-Cr-Sn alloy. We have tried to analysis the Cr-rich phase using the XRD, but the analysis effect is not ideal as the precipitates are small in size. Therefore, we used the EDS.

(5) The study does not mention the number of specimens used in each treatment group.

Re: Thanks for the reminding. For the microhardness and conductivity, each specimen was measured for 5 times to get an average value. For the tensile properties, three samples were tested for each kind of specimens.

(6) the testing temperature used for the tensile tests can be informed in related method section.

Re: Thanks for the reminding. The tensile tests were conducted at 20 ºC in air.

(7) CR3 has the highest mechanical properties. However, in Figure 1, it is clearly seen that CR3 includes more pores than other SEM images. How do the authors explain the contradiction?

Re: Thanks for the reminding. In fact, all the specimens were prepared from the same ingot, and there was no pore before surface corrosion of the specimens. As high density dislocation were introduced into the Cu alloy during the cold rolling process, which promotes the corrosion, and the rolling ratio of CR3 was high, more corrosion pores can be observed on the CR3.

Reviewer 3 Report

Can you compare your method of  cold rolling with industrial?

The possibility of introducing your alloy into industrial processing?

Is there a higher resolution microscopy and EDX analysis to look at the segregation of cations?

Author Response

Can you compare your method of cold rolling with industrial?

Re: Now the high-strength and high-conductivity Cu alloys are usually cold rolled to a certain thickness, and then solid solution treated and aged. This study aims to explore a better processing method, which is a little more complex but can still be conducted using the existing Cu alloy production line.

The possibility of introducing your alloy into industrial processing?

Re: The production process of the Cu-Cr-Sn alloy in this study is relatively simple compared with that of the Cu-Cr-Zr alloy and can be conducted using the conventional Cu alloy production line. Therefore, it is possible.

Is there a higher resolution microscopy and EDX analysis to look at the segregation of cations?

Re: Thanks for the suggestion. All the figures have been replaced by the high resolution figures.
